# YOLO-APDM: Improved YOLOv8 for Road Target Detection in Infrared Images

**DOI:** 10.3390/s24227197

**Published:** 2024-11-10

**Authors:** Song Ling, Xianggong Hong, Yongchao Liu

**Affiliations:** School of Information Engineering, Nanchang University, Nanchang 330019, China; 416100220079@email.ncu.edu.cn (S.L.); 416100220239@email.ncu.edu.cn (Y.L.)

**Keywords:** YOLOv8, infrared road detection, feature fusion, deformable convolution, attention mechanism

## Abstract

A new algorithm called YOLO-APDM is proposed to address low quality and multi-scale target detection issues in infrared road scenes. The method reconstructs the neck section of the algorithm using the multi-scale attentional feature fusion idea. Based on this reconstruction, the P2 detection layer is established, which optimizes network structure, enhances multi-scale feature fusion performance, and expands the detection network’s capacity for multi-scale complicated targets. Replacing YOLOv8’s C2f module with C2f-DCNv3 increases the network’s ability to focus on the target region while lowering the amount of model parameters. The MSCA mechanism is added after the backbone’s SPPF module to improve the model’s detection performance by directing the network’s detection resources to the major road target detection zone. Experimental results show that on the FLIR_ADAS_v2 dataset retaining eight main categories, using YOLO-APDM compared to YOLOv8n, mAP_@0.5_ and mAP_@0.5:0.95_ increased by 6.6% and 5.0%, respectively. On the M3FD dataset, mAP_@0.5_ and mAP_@0.5_ increased by 8.1% and 5.9%, respectively. The number of model parameters and model size were reduced by 8.6% and 4.8%, respectively. The design requirements of the high-precision detection of infrared road targets were achieved while considering the requirements of model complexity control.

## 1. Introduction

As a research hotspot in deep learning, target detection has a wide range of applications in computer vision. In particular, road target detection technology is gradually attracting the attention of relevant departments and companies around the world due to its important value in automated driving and thedynamic planning of road traffic flow [1]. Currently, research in road target identification is primarily focused on the visible light perspective, with few investigations on the infrared perspective. Compared to visible light, infrared radiation can penetrate obstructions such as smoke, fog, and plastic, making infrared imagery advantageous in visual obstructions or bad weather. However, the clarity and resolution of infrared images are lower and the detail is far less than that of visible light images [2,3]. In order to cope with the above problems, reducing background interference and ensuring sufficient information extraction are usually taken as the main research directions in the field of infrared target detection. Examples include sparse representation [4], spatial filtering [5], and frequency domain filtering [6] for target detection in infrared images.

The traditional design concept for infrared target detection is to reduce the target’s background and noise while improving its features. Zhao et al. [5] first used a spatial filtering-based approach to detect infrared targets through background suppression. Anju et al. [6] based on the difference between the target and background frequencies, so as to extract different parts of the frequency to achieve detection. In general, traditional infrared target detection methods use manually designed feature extractors to extract local features using sliding windows, and then use support vector machines to evaluate the detected targets [7]. However, these algorithms suffer from high computational complexity, window redundancy, and poor robustness to multi-scale targets. To improve infrared target identification, Chen et al. [8] created a background suppression module that enhances foreground characteristics. Li et al. [9] presented the YOLO-CAN model, which employs several learning algorithms and enhances the loss function and convolution module to optimize feature extraction in infrared pictures. Zhou et al. [10] introduced the advanced classification network ConvNext and the coordinate attention mechanism, and proposed the channel and spatial attention mechanism for the effect of infrared images, which significantly improves the detection effect of the network on infrared small targets.

Furthermore, since objects in road scenes frequently contain a significant number of regular features, researchers have suggested numerous algorithms for object detection based on feature data. Traditional road target detection approaches use photos to generate artificial feature information based on the target’s attributes, which are subsequently used for target detection. However, traditional detection approaches rely on manually generated feature representations and shallow trainable architectures, and detection performance worsens when these low-level picture features are combined with the target detector or scene classifier’s contextual data [11]. To improve road target detection, Zou et al. [12] introduced an improved SFPN network into the SSD feature extraction network and used ResNet50 instead of VggNet16 [13] as the main feature extraction network for the improved model, further increasing the depth of the network to improve performance. Ma et al. [14] proposed an Improved Small Target Detection (ISOD) network to achieve the fast and efficient detection of small targets by proposing an extended scale feature pyramid network and using an efficient channel attention mechanism for backbone feature extraction. Luo et al. [15] proposed a road small target detection method based on improved YOLOv3, which introduces DIOU Loss [16] to improve the accuracy of localization and optimizes the clustering method in the YOLOv3 algorithm, thus significantly improving the accuracy and speed of detection. Liu et al. [17] thoroughly investigated the model structure and parameter optimization, and proposed the RF-YOLOv3 network model, which was applied to road vehicle detection. The model was developed by a K-means clustering algorithm [18], which determines the number and aspect ratio of target candidate frames based on the unique characteristics of the vehicle, subsequently adjusts the model parameters based on the clustering results, and is used to improve the RF-YOLOv3 network’s detection accuracy and adaptability. Gao et al. [19] designed a road target detection method based on improved YOLOv8n, which significantly improves the detection performance of road targets by fusing the C2f and DBB modules, proposing the PA-AFPN feature fusion method and designing the SPPFT2_TA module.

With the advancement of deep learning and computer vision technologies, many fields have turned to more advanced detection methods that focus on improving detection accuracy, robustness, and adaptability. Convolutional neural network-based target detection methods have become a research hotspot in recent years, and these target detection methods are mainly divided into one-stage regression-based target detection methods and two-stage candidate region-based target detection methods. In 2014, R. Girshick et al. [20] proposed a two-stage target detection algorithm, R-CNN, which was the first time a convolutional neural network was introduced to the field of target detection. Shortly after, R. Girshick et al. [21] further improved R-CNN by introducing SPPNet [22] and proposed Fast R-CNN. Later, Ren, S et al. [23] proposed Faster R-CNN, the first deep learning detection network that can actually be trained end-to-end. These two-stage target detection nets have high computational complexity and significant real-time issues, making them difficult to use on embedded devices with high real-time requirements.

In contrast, one-stage target detection significantly improves detection efficiency by extracting features directly from the image and inferring bounding box location and category confidence. In 2016, Redmon et al. [24] proposed the single-stage target detection algorithm YOLO, which became the most dominant method in target detection with high accuracy and detection speed. YOLO eliminates the need for the region suggestion step by directly detecting all bounding boxes at the same time to unify the object detection step, as well as directly predicting bounding box locations and class probabilities at the image level using deep convolutional neural networks, striking a balance between detection accuracy and performance [25]. Redmon then made a series of improvements and proposed YOLOv2 [26] and YOLOv3 [27], which further improved detection accuracy while maintaining detection speed. In 2020, Bochkovskiy et al. [28] proposed YOLOv4. YOLOv4 experimented with a variety of backbone architectures, culminating in the most powerful backbone network CSPDarknet53, and used a modified version of spatial pyramid pooling from YOLOv3-spp and the same multi-scale predictions as YOLOv3. A few months later, the Ultralytics team presented YOLOv5 [29]. YOLOv5 uses a modified CSPDarknet53 for the backbone and modules such as SCP-PAN and SPPF for the neck. Multiple versions are created by incorporating alternative network depths and widths to fulfill the application and performance needs of various scenarios, considerably boosting the model’s performance, speed, and ease of use. In 2023, the Ultralytics team open-sourced the next major update to YOLOv5, calling it YOLOv8 [30]. YOLOv8 replaces the C3 module in YOLOv5 with the lighter C2f module. YOLOv8 retains the SPPF module while fine-tuning the model at different scales instead of using a single parameter setting, significantly improving model performance. Furthermore, YOLOv8 adds the Anchor-Free Detection header and VFL Loss for classification loss, while combining DFL Loss [31] and CIOU Loss as bounding box loss and Binary Cross Entropy as classification loss. These improvements significantly increase the detection performance and flexibility of the model. As a result, YOLOv8 not only is the first choice for target detection, but also excels in various tasks such as image segmentation and pose estimation.

Infrared road target detection focuses on the labeling and localization of pedestrians and various vehicles, etc., in infrared videos and images. Although the above methods can overachieve good results in road target detection, it is difficult to achieve the expected detection effect for the problem of low image resolution in the infrared view. Aiming at the low-resolution and multi-scale problems of infrared road target detection, this paper proposes an infrared road target detection algorithm, YOLO-APDM, based on YOLOv8. Highly accurate detection of infrared road targets while taking into account the complexity of the control model is achieved.

This paper dedicates the first section to the problem of road target detection in infrared scenes and introduces the research progress of target detection based on convolutional neural networks. Then, Section 2 focuses on the main structure of YOLO-APDM proposed in this paper, and Section 3 details the various improvements of YOLO-APDM. Section 4 introduces the dataset used in this paper and shows the comparison results between YOLO-APDM and the original model. Finally, Section 5 briefly summarizes the work carried out in this paper.

The main contributions of this paper are briefly summarized as follows:Reconstruct the neck of the original model, introduce the P2 layer, optimize the network structure, and improve the multi-scale target detection capability of the model.Improve the C2f module of the model to enhance the ability of the network to focus on the target region and reduce the complexity of the model.Utilize the MSCA mechanism to guide the resources to focus on the most prominent region in the recognition image, thus improving the detection performance of the model.

## 2. The Proposed YOLO-APDM Model

Since its inception as a class of single-target detectors, YOLO has been extensively recognized by academics for its superior detection and real-time performance. In this paper, based on YOLOv8, we perform network structure reconstruction, feature extraction optimization, attention mechanism improvement, and model performance testing before proposing YOLO-APDM, a target detection algorithm for infrared road scenes that achieves high-precision target detection with controlled model complexity.

The overall structure of YOLO-APDM is shown in Figure 1. Similar to the overall structure of YOLOv8, it is mainly composed of backbone, neck, and head. To address the problem of large changes in the scale of road targets, the neck part of the algorithm is improved by using the idea of the fusion of attention scale sequences in ASF-YOLO [32], and the network structure is optimized by introducing the P2 detection layer [33] based on this idea. A detector head for multi-scale target detection is added and integrated with the original predictor head to produce a four-predictor head structure, which improves the network’s multi-scale detection capability. Deformable convolution v3 (DCNv3) [34] is integrated with the C2f module, replacing C2f in the network, and increasing the network’s capacity to focus on the target region while reducing model complexity. At the same time, this paper adopts the multi-scale convolutional attention (MSCA) mechanism in SegNext [35], which allows the model to concentrate its resources on detecting the most important regions in the image, thus improving the model’s detection performance.

## 3. YOLO-APDM Detection Model

### 3.1. Multi-Scale Attention Feature Fusion

The scale of road targets varies widely, ranging from distant automobiles to pedestrians, and even street lighting and markers. To ensure efficient detection performance in such a variable-scale environment, existing algorithms often use a feature pyramid structure for feature fusion. These structures generally sum or concatenate feature maps at different scales. However, existing feature pyramid networks do not completely capitalize on the link between individual pyramid feature maps.

Ming Kang et al. [32] proposed a new YOLO framework named ASF-YOLO in 2024, and the specific structure is shown in Figure 2. ASF-YOLO adopts an attentional scale sequence fusion approach to effectively enhance the multi-scale features extracted from the backbone network by introducing the Scale Sequence Feature Fusion (SSFF) module and Triple Feature Encoder (TFE) module to fuse feature maps at different scales. In light of the aforementioned considerations, the characteristics of the SSFF and TFE modules have been amalgamated through the introduction of a novel Channel and Positional Attention Mechanism (CPAM), which has been designed to optimize the segmentation performance for multi-scale objects in a more efficacious manner. In this paper, we refer to the idea of attentional scale sequence fusion in ASF-YOLO, based on which a P2 detection layer is introduced to optimize the network structure in order to enhance the detection performance of the network in multi-scale feature fusion.

### 3.2. Deformable Convolution Module

Conventional convolutional methods typically employ a fixed-size convolutional kernel for the sampling of input images. This approach is well suited for the processing of targets with fixed shapes, and it can yield satisfactory detection results. However, in practical applications, the shape of the detection target is highly variable, which limits the performance of conventional convolutional methods in adapting to local feature variations. To address this challenge, more convolutional kernels are typically incorporated to extract more complex features. However, this approach not only increases the number of network parameters, but may also affect the feature extraction due to the accumulation of cascading convolutional effects. In this paper, the C2f-DCNv3 is introduced to replace the C2f module in the YOLOv8 network, specific structure is shown in Figure 3. Deformable convolution approximates more accurately the shape and size variations of an object by introducing an offset variable at the sampling position of each convolution kernel and enhances the network’s ability to focus on the target region through a modulation mechanism.

In comparison to DCNv2, DCNv3 exhibits several notable enhancements. Firstly, with regard to the issue of separable convolution, it should be noted that DCNv3 implements shared convolutional weights, denoted as wg. The original convolutional weights are split into depth and point-by-point parts, which improves the model’s efficiency on bigger scales. Secondly, the network’s applicability and flexibility in addressing features are improved by the introduction of the multi-grouping technique. The ability to perform this process with different aggregation patterns at different places on a given layer makes it easier to generate stronger features, which in turn enhances the efficacy of subsequent tasks. Finally, through the application of softmax normalization to the modulation scalar at the sampling points, the stability of the model training at varying scales is enhanced, thereby facilitating the optimization of network training stability. The formula for DCNv3 is shown in Equation (1):(1)$$y(p_{0})=\sum_{g=1}^{G}\sum_{k=1}^{K}w_{g}m_{gk}x_{g}(p_{0}+p_{k}+\Delta p_{k})$$
where G denotes the number of aggregation groups, K denotes the total number of sampling points, and $P_{0}$ denotes the position of the current pixel. The projection weight $w_{g}\in R^{C\times C^{\prime }}$ of each group is independent of the position, where $C^{\prime }=C/G$ denotes the dimension of each group. $m_{gk}\in R$ denotes the modulation of the k−th sampled point of the g−th group after normalization. $p_{k}$ is the predefined network sampled at the kth position and ${∆p}_{k}$ denotes the offset of the k−th grid sampling position.

### 3.3. Multi-Scale Convolutional Attention Mechanism

The attention mechanism, as an adaptive selection process, assists the detection network in focusing its attention on the most relevant sections, so effectively utilizing the model’s computational resources and attention to increase detection performance. Its primary purpose is to focus model effort on crucial areas such as target attributes or background, hence enhancing detection accuracy and robustness. In multiscale processing, the attention mechanism can adjust the importance of features at different scales to effectively capture the details of targets of different sizes, while also improving the model’s ability to deal with occlusions, complex backgrounds, and small targets, allowing it to better adapt to complex real-world scenarios. However, in infrared road target detection, if the target diversity features are extracted through different convolutional channels, it may lead to too much wastage of model resources in non-critical parts of the image. In 2022, Guo et al. [35] created a new multi-scale convolutional attention module (MSCA) in the proposed SegNext, which can achieve the fusion of channel and spatial information, as illustrated in Figure 4.

In particular, MSCA comprises three distinct components: a deep convolutional aggregation of local information, a multi-branch deep strip convolution to capture multi-scale context, and a 1×1 convolution to model the relationship between different channels. The formulas for MSCA are shown in Equations (2) and (3).
(2)$$A\mathrm{tt}=Conv_{1\times 1}(\sum_{i=0}^{3}Scale_{i}(DW-Conv(F))),$$
(3)Out=Att⊗F.
where F are the input features and Att denotes the graph of attention. DW-Conv denotes deep convolution, and ${Scale}_{i},i\in (0,1,2,3)$ is the i−th branch of MSCA, where ${Scale}_{0}$ denotes that the connection is constant. Given the simplicity and intuitiveness of strip convolution, as illustrated in reference [36], it is possible to model deep convolution with a big convolution kernel by using two deep strip convolutions in each branch.

In this paper, the attention mechanism is introduced into the connection between the encoder and decoder by adding the MSCA mechanism after the network SPPF, which helps the model to make better use of the global context information. The MSCA mechanism enables the model to focus more accurately on the region of interest by learning the pixel-level attention weights, while ignoring the background region, which enables the detection network to focus its resources on the region where the main target of the road is detected, and thus improves the detection performance of the model.

## 4. Experiments and Results

### 4.1. Datasets

This study evaluates the current mainstream infrared image datasets based on the FLIR_ADAS_v2 and M3FD datasets. The FLIR_ADAS_v2 dataset [37] is the latest version of the infrared imagery datasets released by FLIR in January 2022, specifically designed for the road environment. Compared with the previous version, these datasets not only cover more categories and richer image content, but also provide a simultaneous annotation of thermal infrared images and unannotated RGB images. The partial images in different scenarios of the FLIR_ADAS_v2 dataset is shown in Figure 5. 

The infrared images of the FLIR_ADAS_v2 dataset consist of 10,742 train images, 3749 test images, and 1144 validation images. They mainly collect 15 types of targets such as people, various vehicles, lighting, and signal lights on the road. This paper selects the eight main categories from the dataset to make the detection more in line with the actual situation of road targets. The specific results are shown in Table 1.

In 2022; Liu et al. [38] created a multi-modality multi-scene dataset M3FD using a simultaneous imaging system which contains 4177 sets of aligned infrared and RGB images and 23,635 samples with object annotations. The M3FD covers a wide range of scenes in multiple environments and under different lighting, seasonal, and weather conditions with a wide range of pixel variations. The dataset is labeled with six common road driving target categories, namely, people, cars, buses, motorbikes, trucks, and lights, which provides strong support for the study of infrared and visible image fusion under extreme conditions. The partial images of the M3FD dataset is shown in Figure 6.

### 4.2. Experimental Indicators

In order to verify the detection performance of the proposed model, this paper uses the average precision value mAP_@0.5_ when the IoU threshold is 0.5 and the average precision value mAP_@0.5:0.95_ in the range of IoU from 0.5 to 0.95 as model performance evaluation indicators to measure the accuracy of model target detection, and combines the parameters, model size, and FLOPs as the evaluation metrics of model complexity. The above metrics can effectively reflect the performance level of the model in terms of detection accuracy and complexity, and the formulae for the relevant metrics are shown in Equations (4) and (5):(4)$$AP=\frac{TP+TN}{TP+TN+FP+FN}$$
(5)$$mAP=\frac{\sum_{n=1}^{Num(classes)}AP(n)}{TP+TN+FP+FN}$$
where TP is the number of positive samples identified correctly; TN is the number of negative samples identified correctly; FP is the number of negative samples incorrectly identified as positive samples; FN is the number of positive samples incorrectly identified as negative samples.

### 4.3. Experiments and Analysis of Results

#### 4.3.1. Experimental Environment

The experiment ran on Windows 10 22H2 operating system with NVIDIA 3060 12 G graphics card as GPU and two Intel(R) Xeon(R) CPU E5-2680 v4 @ 2.40 GHz 2.40 GHz. The deep learning framework used was PyTorch 1.13.1 and training was accelerated by CUDA 11.7. The size of the input image for the training set was kept as 640×640. Stochastic Gradient Descent (SGD) was used as the optimization function and turned off the mixed precision training amp. The version of the network parameters used for training was YOLOv8n (this version is used by default in all the subsequent sections), the epochs were set to 200 rounds, the batch and workers were 16 and 4, and the initial learning rate was 0.01.

#### 4.3.2. Network Optimization Experiments

In order to verify the improvement in detection effect and computational complexity after using ASF-P2 reconstruction in this paper, the original algorithm YOLOv8n was compared with the improved YOLOv8n-ASF-P2 algorithm using ASF-P2, and the comparison results are shown in Table 2.

From the results in Table 2, it can be seen that the improved YOLOv8n-ASF-P2 algorithm for the backbone network is significantly better in detection accuracy compared with the original version. On the FLIR_ADAS_v2 and M3FD datasets, the mAP_@0.5_ values increased by 5.3% and 5.9%, respectively, which verifies that the multi-scale attention feature fusion capability can effectively solve the problem of detection performance degradation caused by the multi-scale characteristics of road images.

#### 4.3.3. Deformable Convolution Module Verification Experiments

Table 3 represents the results after using C2f-DCNv3 to replace C2f in the original network. From the results, it can be seen that the improved model significantly improved all the detection performance indexes. After using the C2f-DCNv3 module, the mAP_@0.5_ of FLIR_ADAS_v2 and M3FD increased by 2.0% and 1.4%, respectively, compared with the original model, and the number of parameters and GFLOPs were reduced by 11.6% and 10.9%, respectively. Thus, it was verified that the use of C2f-DCNv3 not only improves the ability of the network to focus on the target area, and the detection can be more concentrated on the main target, but also effectively reduces the number of parameters of the network and the complexity of the model.

#### 4.3.4. Comparative Experiments on Attention Mechanisms

In order to compare the advantages of the MSCA mechanism in this paper with other attention mechanisms, the common attention mechanism modules SimAM [39], CPCA [40], MLCA [41], CAFM [42], DAttention [43], and LocalWindowAttention [44] were added after the SPPF of YOLOv8n to compare with the attention mechanism used in this paper, with a focus on the changes in detection accuracy. The results are shown in Table 4 and Table 5.

It can be seen from Table 4 and Table 5 that the MSCA mechanism used in this paper significantly improved mAP_@0.5_ and mAP_@0.5:0.95_ compared with other common attention mechanisms. In the infrared road scene detected in this paper, the MSCA mechanism could effectively improve the performance of model detection.

#### 4.3.5. Ablation Experiments

In order to better demonstrate the impact of various improvements in the target detection algorithm on the detection effect, this paper conducted ablation experiments on the various improvements of the model based on YOLOv8n. Table 6 and Table 7 show the results of the ablation experiments on the two datasets, FLIR_ADAS_v2 and M3FD, with the gradual addition of various improvements.

The results in Table 6 and Table 7 show that each improvement of YOLO-APDM could effectively improve the detection performance of the model. Specifically, on the FLIR_ADAS_v2 and M3FD datasets, compared with the original algorithm, YOLO-APDM improved mAP_@0.5_ by 6.6% and 8.1%, respectively, and increased mAP_@0.5:0.95_ by 5.0% and 5.9%, respectively. The number of model parameters and model size were reduced by 8.6% and 4.8%, respectively. However, the GFLOPs increased by 1.5 and there was a slight decrease in FPS due to the increased model layers after the model network reconfiguration; this is still acceptable relative to the significant increase in detection accuracy. The above results further validate the effectiveness of the various improvements of the YOLO-APDM proposed in this paper on the original YOLOv8n model, and prove the superiority of the algorithm proposed in this paper.

In order to demonstrate the improved detection performance of the YOLO-APDM model proposed in this paper in a more intuitive manner, the change curves of the training accuracy of the YOLO-APDM model and several of its improvement terms in comparison with the original YOLOv8n model on the two datasets, FLIR_ADAS_v2 and M3FD, are presented in Figure 7 and Figure 8. These figures illustrate the performance of the models on the two datasets, FLIR_ADAS_v2 and M3FD. As illustrated in the image, the training accuracy of YOLO-APDM on the FLIR_ADAS_v2 dataset is significantly higher than that of the original model from the start of training, and this advantage in accuracy is maintained throughout the succeeding training phases. On the M3FD dataset, the YOLO-APDM model demonstrates superior training accuracy compared to the original model after approximately 40 training rounds. The two results show that the YOLO-APDM model significantly improves training accuracy when compared to the original model.

### 4.4. Comparative Experiments

In order to ascertain the veracity of the assertions made in this paper regarding the excellence of the YOLO-APDM model, a comparison was made between this model and the current mainstream target detection models, including YOLOv3, YOLOv5, YOLOv7, YOLOv8, YOLOv9, YOLOv10, Faster RCNN (using backbone network as ResNet101), Sparse RCNN [45], DETR [46], and RT-DETR [47] (all three using the backbone network as ResNet50). The training was carried out on both the FLIR_ADAS_v2 and M3FD datasets while keeping the individual training parameters and hyperparameters consistent, and the training results are shown in Table 8 and Table 9. The results demonstrate that YOLO-APDM was capable of attaining the optimal equilibrium between model complexity control and model detection performance enhancement in comparison to the aforementioned models.

### 4.5. Visualization and Analysis of Test Results

In order to evaluate the effectiveness and generalization of the YOLO-APDM model, Figure 9 shows the detection results of YOLOv8 and YOLO-APDM on the acquired test images. The results show that YOLO-APDM could effectively detect some of the targets that were difficult to detect in the original model, thus verifying that the improvements of YOLO-APDM can effectively solve the detection problems caused by the low-resolution and multi-scale targets in the infrared road images, and can more accurately detect the targets on the road and improve the overall detection performance.

## 5. Conclusions

In order to solve the problems of low image resolution and multi-scale road targets in infrared road target detection, this paper proposes an improved high-precision infrared road target detection model, YOLO-APDM, based on YOLOv8n, with the design goal of improving the target detection accuracy on the basis of controlling the model parameters. Specifically, this paper adopts the idea of attention scale sequence fusion in ASF-YOLO to reconstruct the neck part of the model pair, and then introduces the P2 detection layer to form a four-prediction head structure, which effectively improves the detection performance of multi-scale targets on the road. In addition, by replacing the deformable convolution v3 module, not only can the number of parameters of the model network be reduced, but the network’s ability to focus on the target area can also be enhanced, thereby improving the network’s flexibility and adaptability. By adding the MSCA mechanism, the detection network resources are concentrated on the key areas of road targets, thereby improving the accuracy and robustness of model detection. Compared with YOLOv8n, YOLO-APDM has significant improvements in major indicators. On the FLIR_ADAS_v2 dataset that retains the main road targets, YOLO-APDM improves mAP_@0.5_ and mAP_@0.5:0.95_ by 6.6% and 5.0%, respectively. On the M3FD dataset, mAP_@0.5_ and mAP_@0.5:0.95_ increased by 8.1% and 5.9%, respectively. The number of model parameters and model size were reduced by 8.6% and 4.8%, respectively. The improved method proposed in this paper achieves higher detection accuracy, while also effectively reducing the number of model parameters and model size, which is conducive to subsequent deployment on embedded devices.

## Figures and Tables

**Figure 1 sensors-24-07197-f001:**
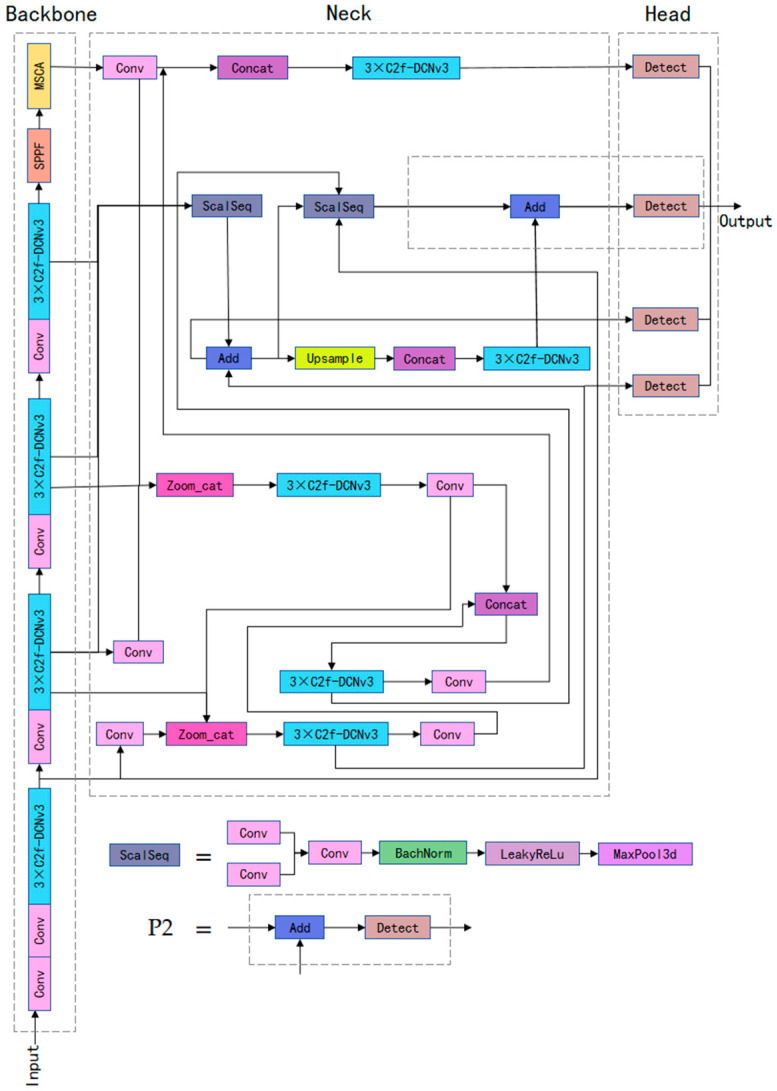
The network structure of YOLO-APDM.

**Figure 2 sensors-24-07197-f002:**
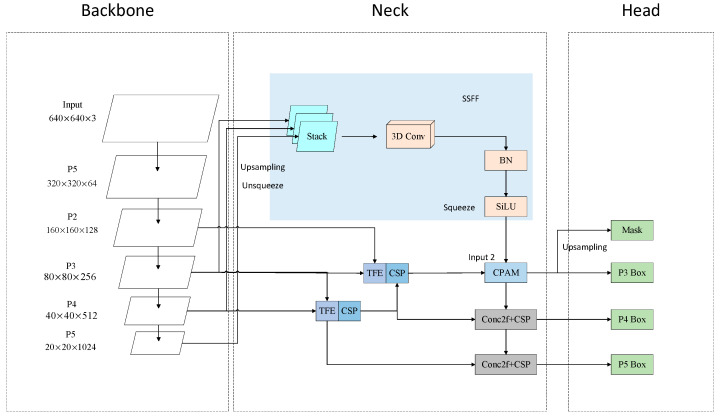
ASF-YOLO structure. The ASF-YOLO framework consists of several key modules, including SSFF, TFE, and Channel and Position Attention Mechanism (CPAM) based on the CSPDarkNet backbone and the YOLO header.

**Figure 3 sensors-24-07197-f003:**
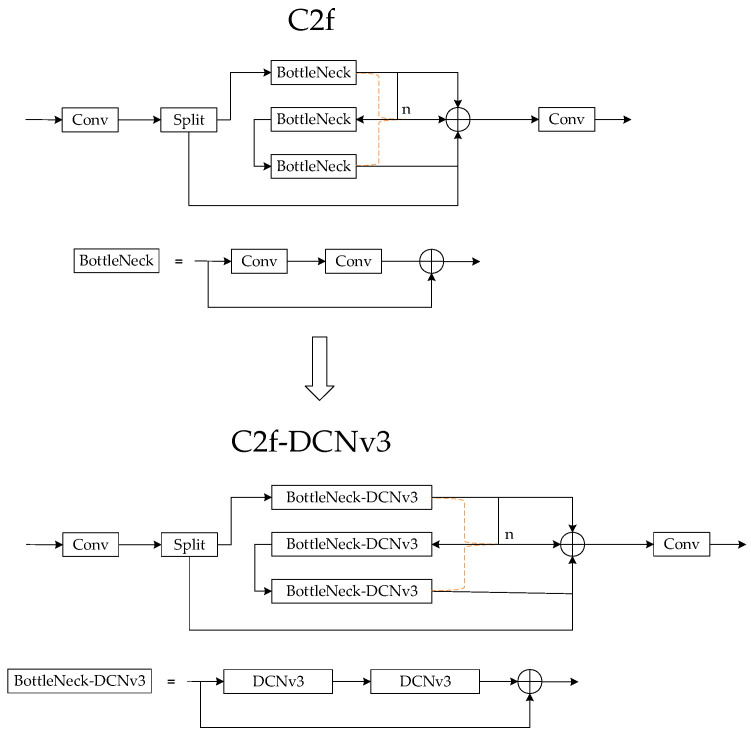
Schematic diagram of C2f-DCNv3. Compared with C2f, the improvement is that the ordinary convolution in the BottleNeck module is replaced by deformable convolution.

**Figure 4 sensors-24-07197-f004:**
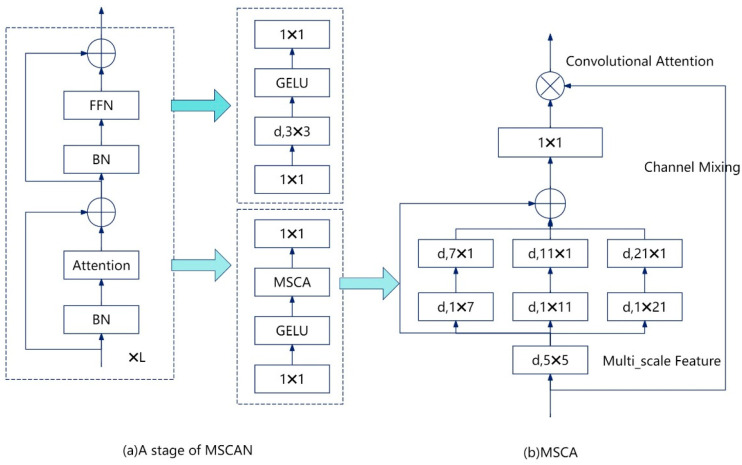
Detailed description of MSCA and MSCAN. In the figure, d denotes the use of deep convolution with kernel size k1 × k2. Multi-scale features are first extracted by the convolution, and then these features are used as attentional weights to re-weigh the inputs to the MSCA.

**Figure 5 sensors-24-07197-f005:**
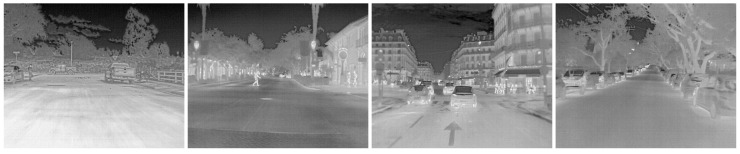
IR maps of the FLIR_ADAS_v2 dataset for different scenarios. This includes daytime, night, occlusion, complex backgrounds, dense, and multi-scale situations.

**Figure 6 sensors-24-07197-f006:**
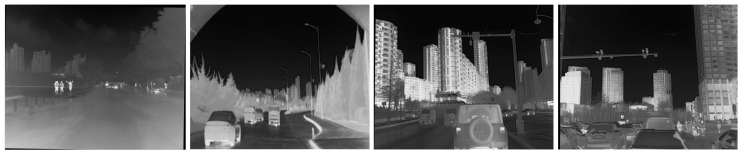
M3FD dataset. This includes six common road driving target categories in different light, seasonal, and weather conditions.

**Figure 7 sensors-24-07197-f007:**
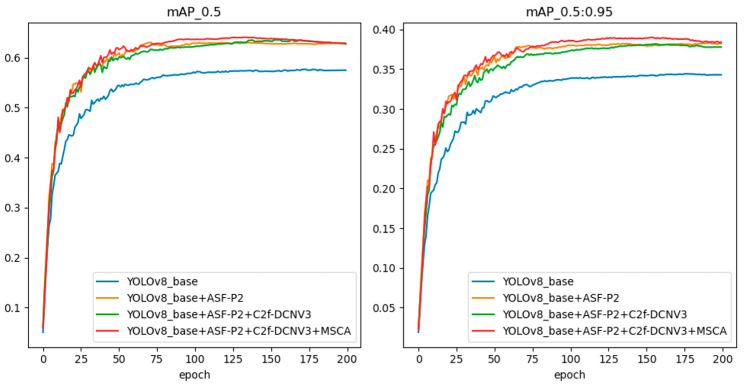
Comparison curves of model training accuracy on the FLIR_ADAS_v2 dataset.

**Figure 8 sensors-24-07197-f008:**
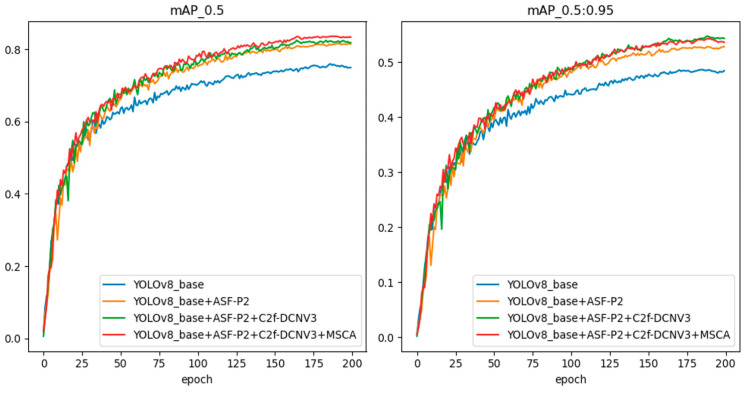
Comparison curves of model training accuracy on the M3FD dataset.

**Figure 9 sensors-24-07197-f009:**
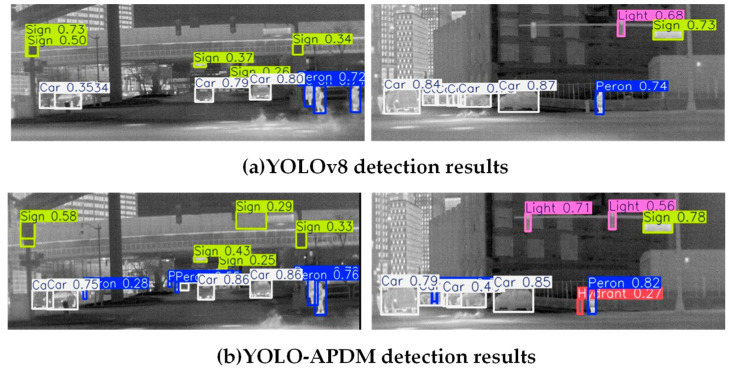
Comparison of detection results.

**Table 1 sensors-24-07197-t001:** Various labels and filtering in the FLIR_ADAS_v2 dataset.

Labels	Train Set Count	Test Set Count	Val Set Count	Reserve Category
Person	50,478	12,323	4470	√
Bike	7237	113	170	√
Car	73,623	30,517	7133	√
Motor	1116	3314	55	√
Bus	2245	0	179	√
Train	5	0	0	
Truck	829	2634	46	
Light	16,198	6758	2005	√
Hydrant	1095	277	94	√
Sign	20,770	5660	2472	√
Dog	4	25	0	
Deer	8	0	0	
Skateboard	29	0	3	
Stroller	15	0	6	
Scooter	15	0	0	
Others	1373	696	63	

**Table 2 sensors-24-07197-t002:** Comparison of detection performance of reconfigured networks using ASF-P2.

Dataset	FLIR_ADAS_v2		M3FD	
	YOLOv8n	YOLOv8n-ASF-P2	YOLOv8n	YOLOv8n-ASF-P2
mAP_@0.5_	0.573	0.626	0.756	0.815
mAP_@0.5:0.95_	0.341	0.377	0.485	0.528
Params (M)	3.01	2.49	3.01	2.49
FLOPs (G)	8.2	12.0	8.2	12.0
Model (MB)	6.3	5.4	6.3	5.4

**Table 3 sensors-24-07197-t003:** Comparative experimental results after replacing the C2f-DCNv3 module.

Dataset	FLIR_ADAS_v2		M3FD	
	YOLOv8n	YOLOv8n-C2f-DCNv3	YOLOv8n	YOLOv8n-C2f-DCNv3
mAP_@0.5_	0.573	0.593	0.756	0.770
mAP_@0.5:0.95_	0.341	0.356	0.485	0.508
Params (M)	3.01	2.66	3.01	2.66
FLOPs (G)	8.2	7.3	8.2	7.3
Model (MB)	6.3	5.6	6.3	5.6

**Table 4 sensors-24-07197-t004:** Comparison results of different attention mechanisms on the FLIR_ADAS_v2 dataset.

Model Name	mAP_@0.5_	mAP_@0.5:0.95_	Params (M)	FLOPs (G)	Model (MB)
YOLOv8n	0.573	0.341	3.01	8.2	6.3
Ours (MSCA)	0.592	0.355	3.00	8.1	6.3
SimAM	0.587	0.351	3.01	8.1	6.3
CPCA	0.577	0.344	3.13	8.3	6.5
MLCA	0.574	0.341	3.11	8.2	6.4
CAFM	0.570	0.340	3.35	8.4	6.9
DAttention	0.577	0.343	3.27	8.3	6.8
LocalWindowAttention	0.576	0.343	3.10	8.2	6.5

**Table 5 sensors-24-07197-t005:** Comparison results of different attention mechanisms on the M3FD dataset.

Model Name	mAP_@0.5_	mAP_@0.5:0.95_	Params (M)	FLOPS (G)	Model (MB)
YOLOv8n	0.756	0.485	3.01	8.2	6.3
Ours (MSCA)	0.775	0.503	3.00	8.1	6.3
SimAM	0.768	0.497	3.01	8.1	6.3
CPCA	0.766	0.493	3.13	8.3	6.5
MLCA	0.752	0.484	3.11	8.2	6.4
CAFM	0.744	0.489	3.35	8.4	6.9
DAttention	0.764	0.502	3.27	8.3	6.8
LocalWindowAttention	0.758	0.499	3.10	8.2	6.5

**Table 6 sensors-24-07197-t006:** Comparison results of ablation experiments on the FLIR_ADAS_v2 dataset.

v8n_Base	ASF-P2	DCNv3	MSCA	mAP_@0.5_	mAP_@0.5:0.95_	P (%)	R (%)	Params (M)	FLOPs (G)	FPS	Model (MB)
√				0.573	0.341	0.739	0.500	3.01	8.1	33.9	6.3
√	√			0.626	0.377	0.734	0.545	2.49	12.0	24.0	5.4
√		√		0.593	0.356	0.716	0.526	2.66	7.3	37.8	5.6
√			√	0.592	0.355	0.731	0.523	3.00	8.1	49.7	6.3
√	√	√		0.631	0.382	0.774	0.524	2.67	9.7	27.7	5.9
√	√	√	√	0.639	0.391	0.733	0.531	2.75	9.6	30.2	6.0

**Table 7 sensors-24-07197-t007:** Comparative results of ablation experiments on the M3FD dataset.

v8n_Base	ASF-P2	DCNv3	MSCA	mAP_@0.5_	mAP_@0.5:0.95_	P (%)	R (%)	Params (M)	FLOPs (G)	FPS	Model (MB)
√				0.756	0.485	0.823	0.682	3.01	8.1	34.6	6.3
√	√			0.815	0.528	0.859	0.721	2.49	12.0	26.1	5.4
√		√		0.770	0.508	0.856	0.671	2.66	7.3	38.6	5.6
√			√	0.775	0.503	0.818	0.698	3.00	8.1	46.1	6.3
√	√	√		0.820	0.537	0.850	0.751	2.67	9.7	27.5	5.9
√	√	√	√	0.837	0.544	0.867	0.763	2.75	9.6	30.8	6.0

**Table 8 sensors-24-07197-t008:** Comparison of training results of different models on the FLIR_ADAS_v2 dataset.

Model Name	mAP_@0.5_	Params (M)	FLOPs (G)	Model (MB)
YOLOv3-tiny	0.557	12.12	19.0	24.5
YOLOv5n	0.541	1.77	4.3	3.9
YOLOv7-tiny	0.599	6.03	13.2	12.3
YOLOv8n	0.573	3.01	8.2	6.3
YOLOv9t	0.585	1.97	7.6	4.6
YOLOv10n	0.578	2.27	6.5	5.7
Faster RCNN (R101)	0.525	42	-	232
Sparse RCNN(R50)	0.582	27	-	406.1
DETR (R50)	0.532	41	-	159
RT-DETR(R50)	0.570	42	-	164
Ours (YOLO-APDM)	0.639	2.75	10.3	6.0

**Table 9 sensors-24-07197-t009:** Comparison of training results of different models on the M3FD dataset.

Model Name	mAP_@0.5_	Params (M)	FLOPs (G)	Model (MB)
YOLOv3-tiny	0.716	12.12	19.0	24.5
YOLOv5n	0.715	1.77	4.3	3.9
YOLOv7-tiny	0.741	6.03	13.2	12.3
YOLOv8n	0.756	3.01	8.2	6.3
YOLOv9t	0.758	1.97	7.6	4.6
YOLOv10n	0.764	2.27	6.5	5.7
Faster RCNN (R101)	0.688	42	-	232
Sparse RCNN (R50)	0.744	27	-	406.1
DETR (R50)	0.696	41	-	159
RT-DETR (R50)	0.748	42	-	164
Ours (YOLO-APDM)	0.837	2.75	10.3	6.0

## Data Availability

FLIR_DADAS-v2 Dataset: The FLIR_DADAS-v2 dataset used in this study is publicly available and can be accessed from the FLIR official website: https://www.flir.in/oem/adas/adas-dataset-form (accessed on 27 August 2024). M3FD Dataset: The M3FD dataset used in this study is publicly available and can be accessed from the official GitHub repository: https://github.com/dlut-dimt/TarDAL (accessed on 27 August 2024).

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
