# Peer review of "YOLO-APDM: Improved YOLOv8 for Road Target Detection in Infrared Images"

_sensors, 2024, doi:10.3390/s24227197_

Round 1

Reviewer 1 Report

Comments and Suggestions for Authors

This paper proposes an improved YOLOv8 for road target detection in infrared images. In general, the proposed method sounds interesting and some comments are given as follows:

1. The introduction should give a brief summary of each chapter to aid readers in understanding the paper's structure. Adding such an overview will enhance the paper's organization and readability.   2. The boxes of each module in Figure 1 are too small, which makes the overall image unclear and less aesthetically pleasing. It is recommended to optimize the layout by enlarging the size of the boxes in each module to ensure the text and content  are clearly visible. Additionally, using a high-definition vector graphic format is suggested to improve the resolution and maintain clarity.   3. The content of the second chapter is too brief, with only one paragraph of text and one image. It is recommended to adjust the structure of the article and expand the content of this chapter.   4. The "where" following the formula does not need to be in a separate paragraph. Additionally, the line spacing for introducing parameters below the formula is not appropriate, making the formatting look irregular. It is recommended to adjust the format to ensure the document is neat and well-organized.   5. All the results of the article are based on two public datasets. To verify the effectiveness of the algorithm, it is recommended to validate it on photos obtained from your own experiments to enhance the credibility and practicality of the experimental results.   6. Other related YOLO application senarios are suggested to be discussed in the literature review part, e.g., small object, ISOD: Improved small object detection based on extended scale feature pyramid network, Visual Computer, 2024.

Author Response

请参阅附件

Reviewer 2 Report

Comments and Suggestions for Authors

This paper proposes a new algorithm called YOLO-APDM to address low-quality and multi-scale target detection issues in infrared road scenes. However, some comments may improve the quality of the manuscript such as the following:

1. The abstract section should include the manuscript's novelty problems.

2. There is still room to add some important relevant references that are missing. See for example the following works: Improved road object detection algorithm for YOLOv8n, Computer Engineering and Application, 2024, 60(16): 186-197; Phase diagram in multi-phase heterogeneous traffic flow model integrating the perceptual range difference under human-driven and connected vehicles environment, Chaos, Solitons & Fractals, 2024,182: 114791;

3. The introduction section must be enhanced to present the main concerns of the present study; the authors should look what is the difference between the present and the previous.

4. The description of the main contributions is brief and not specific enough. To facilitate readers to recognize the innovation of this manuscript quickly, it is suggested to elaborate on the main contributions in several points in the Introduction section.

5. The limitation(s) of this work should be discussed. Other possible methodologies that can be used to achieve the objective relating to this work should also be analyzed.

6. The whole paper needs to be rechecked as there are lot many grammatical and typos errors.  

Comments on the Quality of English Language

The whole paper needs to be rechecked as there are lot many grammatical and typos errors.  

Reviewer 3 Report

Comments and Suggestions for Authors

Based on YOLOv8, the paper proposes YOLO-APDM to address low quality and multi-scale 8 target detection issues in infrared road scenes. Experiments on FLIR_ADAS_v2 and M3FD datasets verified the effectiveness of YOLO-APDM.

However, it seems that the paper is much more application-oriented rather than methodology-oriented, which lacks the novelty of techniques. My detailed comments are as follows:

1, The resolution in Figure 1 is low. The correspondence between the main contribution of this paper and the modules in the figure is unclear. The P2 module is not indicated in the figure, and there are no references or specific structures for the MSCA module in the figure.

2, The relationship between Figure 2 and the main contribution of this paper is unclear.

3, It is suggesting the author to supplement the comparison of model inference time.

4, It is suggesting the author to supplement comparative experiments with models outside YOLO series, such as DETR series models.

5, The resolution of Figures 9 and 10 is too low to see clearly.

6, The formula format in the paper does not meet the requirements of the journal.

Round 2

Reviewer 1 Report

Comments and Suggestions for Authors

The authors have addressed all my comments.

Author Response

Thank you for your constructive comments on this article!

Reviewer 2 Report

Comments and Suggestions for Authors

After reviewing the revised manuscript, I found that all the comments responded properly. I give my recommendation to accept this paper.

Author Response

Thank you very much for your valuable comments, which help us to improve the quality of this article more effectively.

Reviewer 3 Report

Comments and Suggestions for Authors

The development of object detection methods is very rapid. I suggest that the author should fully compare the proposed method with influential object detection methods proposed in the past two years in the experiment, in order to demonstrate the advantages of this method. Which means the author should demonstrate the performance of YOLO-APDM in practical applications through experiments, as the method is lacks the novelty of techniques. 

This paper can be accepted subject to addressing the above question.
